# Deep learning for pediatric chest x-ray diagnosis: Repurposing a commercial tool developed for adults

Prerana Agarwal[1]*, Alexander Rau[1], Helen Ngo[1], Ambika Seth[2], Fabian Bamberg[1], Elmar Kotter[1], Jakob Weiss[1]

**1** Department of Diagnostic and Interventional Radiology, Medical Center - University of Freiburg, Freiburg, Germany, **2** Lunit, 7F, 374, Gangnam-daero, Gangnam-gu, Seoul, Korea

* prerana.vkag@gmail.com

## Abstract

The number of commercially available artificial intelligence (AI) tools to support radiological workflows is constantly increasing, yet dedicated solutions for children are largely unavailable. Here, we repurposed an AI-tool developed for chest radiograph interpretation in adults (Lunit INSIGHT CXR) and investigated its diagnostic performance in a real-world pediatric clinical dataset. 958 consecutive frontal chest radiographs of children aged 2−14 years were included and analyzed with the commercially available AI-tool. The reference standard was determined in a dedicated reading session by a board-certified radiologist. The original reports validated by specialized pediatric radiologists, were considered as second readings. All discordant findings were reanalyzed in consensus. The diagnostic performance of the AI-tool was validated using standard measures of accuracy. For this, the continuous AI output (ranging from 0−100) was binarized using vendor recommended thresholds recommended for adults and optimized thresholds identified for children. Relevant findings were defined as consolidation, atelectasis, nodule, cardiomegaly, mediastinal widening due to mass, pleural effusion and pneumothorax. 200 radiographs [20.9%] demonstrated at least one relevant pathology. Using the adult threshold, the AI-tool showed a high performance for all relevant findings with an AUC 0.94 (95% CI: 0.92–0.95) and. In stratified analysis by age (2−7 vs. 7–14-years-old) a significantly higher performance (p < 0.001) was found for older children with an AUC of 0.96 (95% CI: 0.94–0.98) with a sensitivity and specificity of 87.5% and 82.3% respectively, which further increased using optimized thresholds for children. Repurposing existing AI-tools developed for adult application to pediatric patients could support clinical workflows until dedicated solutions become available.

**Data availability statement:** Due to institutional data privacy regulations and patient confidentiality at the University Hospital Freiburg, the data cannot be shared publicly. However, the data may be made available upon reasonable request by contacting the institution at rdia.studienzentrum@uniklinik-freiburg.de.

**Funding:** Our institute received a grant from Lunit for technical support of the study. The funders had no role in study design, data collection and analysis, decision to publish the manuscript. Some information regarding the AI tool (training data) was provided for manuscript preparation.

**Competing interests:** The authors have declared that no competing interests exist.

**Abbreviations:** AI, Artificial Intelligence; AUC, Area under the receiver operating characteristic curve; CI, Confidence Interval; CXR, Chest X-Ray; DL, Deep learning; NPV, Negative predictive value; PPV, Positive predictive value; SD, Standard deviation

## Introduction

Chest radiography remains one of the most commonly used imaging tests in the pediatric population and serves as an important tool in the workup of various conditions involving the lungs, mediastinum and chest wall [1]. Over the past years, the number of commercially available artificial intelligence (AI) tools to support radiological workflows has substantially increased, especially in the field of chest radiography with applications encompassing a wide range of clinical scenarios such as nodule or pneumonia detection, tuberculosis screening, as well as triaging and streamlining workflow [2–6]. However, the focus of this rapidly evolving landscape has remained with the adult population and dedicated solutions for pediatric patients are limited. Among the currently FDA-cleared commercially available AI solutions, there are, to date, no computer-aided detection products specifically authorized for pediatric use [7]. This gap is attributable to several factors, including logistical challenges such as the limited availability of high-quality, open-access pediatric imaging datasets, and the scarcity of pediatric subspecialty radiologists required for accurate image annotation. Additional barriers include stricter regulatory requirements for pediatric research, smaller market potential compared to adult applications, and the inherent variability in clinical conditions and anatomical characteristics associated with the diverse age range within the pediatric population [8–10].

In this context, repurposing existing AI-tools developed for adult application to pediatric patients could enhance clinical workflows, aid in decision-making and support pediatricians in a resource constrained set-up until dedicated solutions become available. In recent years, efforts have been made to adapt adult chest AI algorithms for pediatric use in order to accelerate the development of pediatric imaging AI. Various approaches have been explored, including the exclusion of children under two years of age and the omission of certain findings such as cardiomegaly to improve performance, as well as optimizing the operating thresholds of AI tools [11–13]. Nonetheless, thorough and rigorous testing is mandatory to gain a better understanding of potential benefits and limitations.

Here, we investigated the diagnostic performance of a commercially available AI-tool developed for adult chest radiograph interpretation (Lunit INSIGHT CXR) in a real-world clinical dataset of children aged 2–14 years old. Our hypothesis was that when used for specific relevant pathologies, such as consolidation, pneumothorax or effusion, repurposed AI tools for chest radiograph analysis can reliably detect such pathologies and support clinical management.

## Materials and methods

### Patient population

In this single-center retrospective cohort study to independently externally validate a commercially available AI tool, we included frontal chest radiographs of 1000 consecutive children between 2–14 years old who received a clinically indicated chest

radiograph at our tertiary care center between January 2021 and April 2022 (Fig 1). For the purpose of the study, the data was accessed between 01.02.2023 and 30.06.2023.

Inclusion criteria were chest radiographs acquired in a postero-anterior or antero-posterior projection in the routine clinical workup in children aged 2–14. Children under the age of 2 years were not included in our study since they often show substantially differing anatomical features and a different spectrum of pathologies compared to older children. Since the AI tool is already approved for use in children older than 14 years, we limited the age group for our study from 2–14 years. Exclusion criteria were cases with poor/corrupted image quality such as strong rotation, artifacts due to motion or studies not fully covering the chest.

This study was performed in line with the principles of the Declaration of Helsinki. Approval was granted by the Ethics Committee of University of Freiburg (22–1184-retro) and informed consent was waived.

## Chest radiographs

All chest radiographs were acquired in clinical routine and were retrieved in DICOM format from the Picture Archiving and Communication System (PACS). The images were obtained using different radiography units (Siemens Mobilett XP, Philips Digital Diagnost and Mobile Diagnost and Samsung GM 60A and 85A). Only one chest radiograph per patient was included. After visually checking for impaired/corrupted image quality, the chest radiographs were sent to a dedicated image analysis post-processing platform NORA (www.nora-imaging.com) for further analysis.

**Reference standard reading.** A board-certified radiologist specialized in chest imaging and with experience in pediatric imaging (P.A., 10 years of experience) analyzed the radiographs in a dedicated reading session, in which the

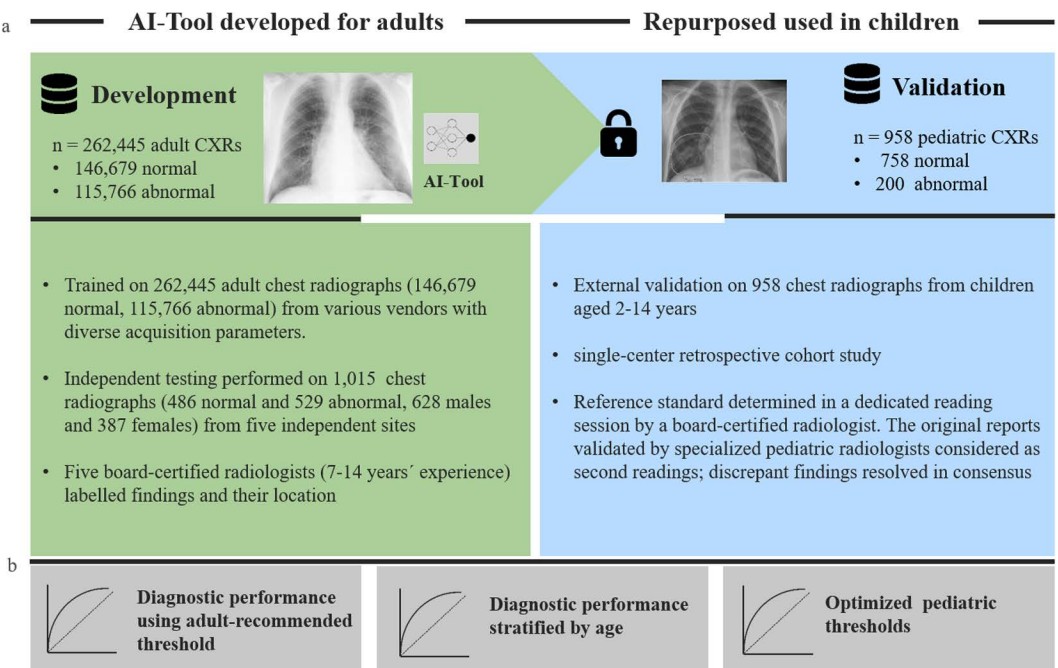

**Fig 1. Brief summary highlighting the study methology. a) AI Tool Development and Pediatric Repurposing: The AI tool was originally trained and validated using a large dataset of adult chest radiographs.** For pediatric validation, the tool was retrospectively tested on 958 pediatric chest radiographs (CXR) from children aged 2–14 years. **b) Diagnostic Performance Analysis:** The AI tool's diagnostic performance in children was assessed using vendor-recommended thresholds, stratified by age groups (2–6 and 7–14 years), and optimized pediatric-specific thresholds.

presence/absence of predefined relevant pathologies was recorded for each chest radiograph and annotated in the image for later comparison and verification of the AI model outputs (Fig 2). Relevant findings were defined as: 1) consolidation, 2) atelectasis, 3) nodule, 4) cardiomegaly, 5) mediastinal widening due to mass, 6) pleural effusion and 7) pneumothorax. To generate the best possible reference reads for the clinical question, in addition to the chest radiograph, all additional information available (e.g., lab results, CT scans, clinical history) was taken into account. The original signed reports, which were validated by specialized pediatric radiologists, were considered as second readings. Finally, if any discordant findings were noted between the dedicated study reads and the original signed reports, the case was reanalyzed in consensus by P.A. and J.W. to determine the final reference standard read for this study.

## AI-Algorithm

The radiographs were evaluated with a commercially available fully automatic AI tool (Lunit INSIGHT CXR, Version 3.1.4.4), which was originally developed for chest radiograph (postero-anterior or antero-posterior projection) interpretation in adults. For training of the AI, 262,445 adult frontal chest radiographs (146,679 normal and 115,766 abnormal) were used from various vendors and with a wide range of acquisition parameters and across a wide range of geographic regions. Independent testing was performed on a total of 486 normal and 529 abnormal chest radiographs collected from five independent institutions (1 from each participant, 628 males and 387 females). Five board-certified radiologists with 7–14 years of experience participated in labeling the chest radiographs, providing the type and the exact location of the abnormalities. Further details on the original development and testing are provided elsewhere in detail [14]. The following ten findings are detected by the algorithm: consolidation, atelectasis, nodule, fibrosis, calcification, pleural effusion, pneumothorax, pneumoperitoneum, cardiomegaly and mediastinal widening. The output is provided as a heatmap or a grayscale map to indicate the location of the finding on the chest radiograph, along with with a probability or abnormality score between 0–100 for each finding, reflecting the certainty of the AI tool (Figs 2 and 3). The software classifies the lesions as positive using a cut-off of 15%, which is the optimized for the adult population.

Thus far, the AI tool has European Clearance with a Medical Device Regulation Certificate (MDR) for individuals >=14-years-old, Korean Food and Drug Administration Clearance (KFDA) and Brazilian ANVISA clearance and has been widely tested in different clinical scenarios in its intended area of use [15–18]. For the current study with a

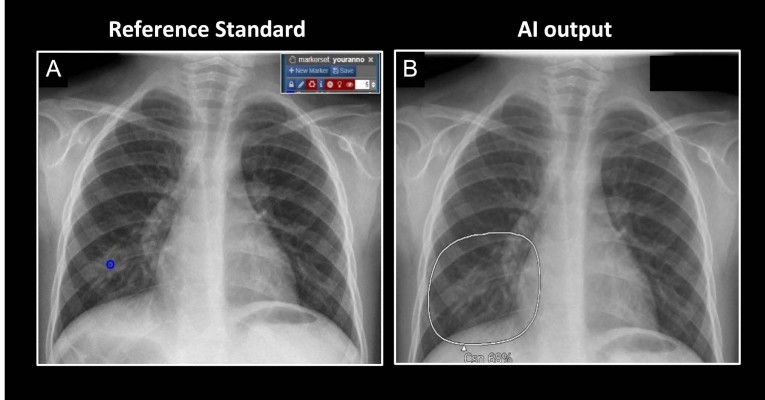

**Fig 2. Example of reference standard and AI output. A)** Reference standard with annotated finding by board-certified radiologist specializing in thoracic imaging. Blue marker indicates a consolidation in the right lower zone. The box in the upper right corner shows the annotation tool of the image-processing platform NORA. **B)** AI output with grayscale map showing consolidation (Csn) with an abnormality score of 68% in the right lower zone, considered as a true positive finding.

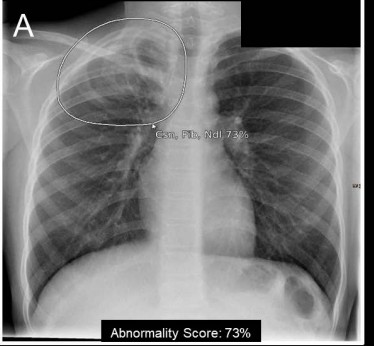 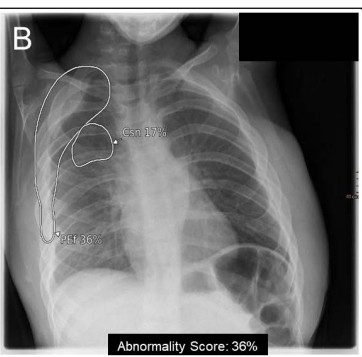

**Fig 3. Image examples representing strengths and limitations of the AI tool.** A) AI output in a 10-year-old patient with a history of Ewing sarcoma of the 1st right rib. The area was highlighted as pathologic with an abnormality score of 73% and classified as consolidation (Csn), fibrosis (Fib) and nodule (Ndl), which most closely resemble the findings the AI tool was developed for. B) Image of a 4-year-old child with a venolymphatic malformation of the chest wall. Similar to **A** an abnormality was correctly detected but erroneously classified as effusion (PEf) and consolidation (Csn) as it was beyond the application of the AI tool.

repurposed application in children, Lunit provided technical support but was not involved in the study design, data collection or data analysis, or decision to publish.

## Performance analysis of the AI tool

All chest radiographs meeting the inclusion criteria were sent to the AI tool for automatic analysis. After analysis, the manually annotated reference chest radiograph and the grayscale map overlaid radiographs generated by the AI tool were re-read by one of the authors (H.N.) to identify potential false positives by the AI tool (same findings but different locations in the image).

## Statistical analysis

The assumption of normal data distribution of the data was tested with the Shapiro–Wilk test. Continuous variables are presented as mean±standard deviation (SD) or median and interquartile ranges (IQR) as appropriate. Categorical variables are given as frequencies and percentages. The diagnostic performance of the AI tool was investigated using the dedicated reference reads generated for this study as gold standard and reported as per the STARD statement [19]. The continuous output of the AI tool was used to calculate the area under the receiver operating characteristic curve (AUC) with 95% confidence intervals (95% CI) for the combined performance across all findings. In addition, sensitivity, specificity, positive predictive value (PPV), negative predictive value (NPV) were calculated after binarizing the continuous AI output using a predefined, vendor recommended threshold cut-off of 15, which was the optimal threshold identified for adults. To account for the relatively small number of individual findings, consolidation, nodule, atelectasis, pleural effusion and pneumothorax were combined to pleuroparenchymal and cardiomegaly and mediastinal widening to mediastinal findings to allow for reasonable statistics. First, performance metrics were calculated for the entire dataset using the vendor recommended threshold recommend for adults. Furthermore, subanalyses stratified by were conducted in a similar fashion to investigate a potential age bias. As a final step, instead of using the predefined vendor recommended threshold, an overall optimal cut-off was calculated followed by separate cut-offs for pleuroparenchymal and mediastinal subgroups by maximizing the sum of sensitivity and specificity (R-Package cutpointr). AUCs were compared using the DeLong method [14]. P-value of less than 0.05 was considered to be statistically significant. All analyses were conducted with R (version 4.3.0; R Foundation for Statistical Computing, Vienna, Austria).

**Table 1. Demographic information.**

| Parameter | Total (n = 958) | Boys (n = 524) | Girls (n = 434) |
|---|---|---|---|
| **Age (years)** [*] | 6.75 ± 3.6 | 6.83 ± 3.6 | 6.65 ± 3.6 |
| **Projection (AP)** | 500/958 (52.2%) | 274/524 (52.3%) | 226/434 (52.1%) |
| **Relevant overall pathology** | 200/958 (20.9%) | 96/524 (18.3%) | 104/434 (23.4%) |
| • Pleuroparenchymal pathology [†] | 162/200 (81.0%) | 83/200 (41.5%) | 79/200 (39.5%) |
| • Mediastinal pathology [‡] | 77/200 (38.5%) | 29/200 (14.5%) | 48/200 (24.0%) |
| • Both mediastinal and pleuroparenchymal pathology | 39/200 (19.5%) | 16/200 (8.0%) | 23/200 (11.5%) |

[*]Data represents mean ± SD with range in brackets

[†]Relevant pleuroparenchymal pathology includes consolidation, nodule, atelectasis, pleural effusion and pneumothorax

[‡]Relevant mediastinal pathology includes cardiomegaly and mediastinal widening due to a mass.

AP = Anteroposterior

## Results

### Patient cohort

Out of the 1000 radiographs, 42 cases were excluded due to poor/corrupted image quality, resulting in a final study cohort of 958 patients (524 boys; 6.75 ± 3.6 years) with 200 radiographs [20.9%] demonstrating at least one relevant pathology (Table 1).

### Algorithm performance

Performance analysis based on the thresholds recommended for adults:

A summary of the AI tool performance is presented in Table 2. The overall performance of the AI tool for identifying any relevant pathology was high with an AUC of 0.94 (95% CI: 0.92–0.95) and an accuracy of 83.4% (95% CI: 80.1–85.7%). The sensitivity, specificity, PPV and NPV were 87.5%, 82.3%, 56.6% and 96.1%, respectively.

For the identification of relevant pleuroparenchymal pathologies (consolidation, nodule, atelectasis, pleural effusion and pneumothorax), the algorithm showed an AUC of 0.94 (95% CI: 0.92–0.96) and an accuracy of 85.3% (95% CI: 82.9–87.5%). The sensitivity and specificity were 87.7% and 84.8%, respectively. PPV and NPV were 54% and 97.1%, respectively.

For mediastinal pathologies (cardiomegaly and mediastinal widening due to a mass), the metrics were as follows: AUC 0.94 (95% CI: 0.92–0.96); accuracy 89% (95% CI: 86.6–90.9%); sensitivity, specificity, PPV and NPV: 72.7%, 90.4%, 40% and 97.4%, respectively.

**Performance analysis stratified by age.** To investigate the impact of age on the algorithm performance, we performed a subgroup analysis between children aged 2–6 years vs. 7–14 years using the corrected performance metrics. This age cut-off was chosen as it allowed for a balanced distribution of patients, allowing for statistically meaningful comparison between early childhood and school-aged children. Furthermore, in a previous study evaluating the same AI tool, 7 years was identified as the median age for correct diagnosis [12].

In the younger age group (n = 509 [53.1%]; mean age 3.7 ± 1.4 years), the AUC for the presence of any relevant finding was 0.91 (95% CI: 0.88–0.94), with an accuracy of 78% (95% CI: 74.1–81.5%) and a sensitivity, specificity, PPV and NPV of 88.1%, 74%, 57% and 94.1%, respectively. In the older age group (n = 449 [46.9%]; mean age 10.2 ± 2 years), the performance metrics were as follows: AUC 0.96 (95% CI: 0.94–0.98), accuracy 89.7% (95% CI: 86.5–92.4%), sensitivity 86.2%, specificity 90.3%, PPV 56.8% and NPV 97.8%, respectively, which were significantly higher compared to the younger age groups (p < 0.001).

 

**Table 2. Results of the algorithm performance: For the calculation of the diagnostic performance of the AI tool, dedicated reference reads generated for this study were used as a gold standard.**

| Pathology | Cases (n=958) | AUC (95%CI) | Accuracy (95% CI) | Sensitivity (%) | Specificity (%) | PPV (%) | NPV (%) |
|---|---|---|---|---|---|---|---|
| **Relevant pathology** | 200 (20.9%) | 0.94 (0.92-0.95) | 83.4 (80.1-85.7) | 87.5 (175/200) | 82.3 (624/758) | 56.6 (175/309) | 96.1 (624/649) |
| **Pleuroparenchymal pathology** | 162 (16.9%) | 0.94 (0.92-0.96) | 85.3 (82.9-87.5) | 87.7 (142/162) | 84.8 (675/796) | 54 (142/263) | 97.1 (675/695) |
| **Mediastinal pathology** | 77 (8%) | 0.94 (0.92-0.96) | 89 (86.9-90.9) | 72.7 (56/77) | 90.5 (797/881) | 40 (56/140) | 97.4 (797/818) |
| **Individual pathologies***  | **Cases** | **True Positives** | **False Negatives** | | **False Positives** | **True Negatives** | |
| Consolidation | 144 (15%) | 84.7% (122/144) | 15.3% (22/144) | | 14.6% (119/814) | 85.4% (695/814) | |
| Atelectasis | 24 (2.6%) | 41.7% (10/24) | 58.3% (14/24) | | 1.7% (16/934) | 98.3% (918/934) | |
| Nodule | 9 (1%) | 88.9% (8/9) | 11.1% (1/9) | | 6.4% (61/949) | 93.6% (888/949) | |
| Pleural effusion | 27 (2.8%) | 66.7% (18/27) | 33.3% (9/27) | | 1.5% (14/931) | 98.5% (917/931) | |
| Pneumothorax | 2 (0.2%) | 100%(2/2) | 0% (0/2) | | 0.1% (1/956) | 99.9% (955/956) | |
| Cardiomegaly | 72 (7.5%) | 73.6% (53/72) | 26.4% (19/72) | | 8.4% (74/886) | 91.6% (812/886) | |
| Mediastinal widening | 6 (0.6%) | 100% (6/6) | 0% (0/6) | | 7.5% (71/952) | 92.5% (881/952) | |

AUC = Area under the receiver operating characteristic curve

PPV = Positive Predictive Value

NPV = Negative Predictive Value

*For individual pathologies, the AI-tool performance is depicted in percentages and raw numbers as standard metric of accuracy could not be reliably calculated due to small sample sizes.

A detailed summary of all findings is presented in S1 and S2 Tables.

**Optimal thresholds for pediatric patients.** The vendor-recommended threshold of 15 to binarize the continuous AI output (ranging from 0–100) is based on the optimal threshold identified for adults. Therefore, as a final step in our analysis, we calculated optimized cut-offs for children for the entire cohort. Overall, the optimal cut-off for the pediatric age group was found to be 44.5. This adjusted threshold yielded a similar AUC of 0.93 with an accuracy of 89.8% (95% CI: 87.7–91.6%), sensitivity of 80.0%, and specificity of 92.4%. Compared to the adult threshold, the pediatric-specific cut-off led to improvement in the specificity and the PPV, supporting a safer AI interpretation in children with a slight trade-off in the sensitivity at 80%. The sub-group specific thresholds for pleuroparenchymal and mediastinal subgroups are shown in the Fig 4 and for each pathology are shown in S1 Fig.

## Discussion

In this study, we repurposed and externally validated the diagnostic performance of a commercially available AI-tool (Lunit INSIGHT CXR) which was originally developed for chest radiograph evaluation for adults in a real-world pediatric dataset. When benchmarked against dedicated reference reads generated for this study, the algorithm exhibited a high and clinically acceptable performance for relevant findings. Notably, subanalyses by age revealed a significantly higher performance for older children aged 7–14 years, compared to younger patients (2–6 years), presumably due to a substantially different anatomy in younger individuals. For example, the presence of a thymic shadow could pose a challenge, resulting in its erroneous identification as cardiomegaly or mediastinal enlargement. Additionally, perhaps the distinctly visible vascular patterns in the lungs of supine pediatric subjects were occasionally misinterpreted as pulmonary edema (the algorithm classifies pulmonary edema as consolidation).

Our results are of clinical importance, because most research on AI tools for pediatric chest radiograph interpretation over the past years focused on pneumonia detection and automatic segmentation of the lungs while comprehensive solutions evaluating more pathologies are still missing [20–22]. The concept of repurposing AI tools developed for the adult

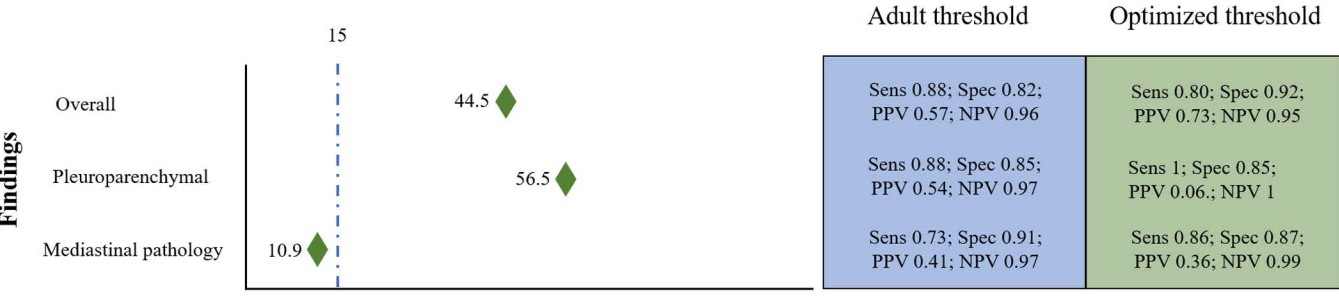

**Fig 4. Definition of optimal thresholds for the performance of the AI-tool in children.** The dotted blue line represents the pre-defined vendor recommended threshold of 15, which is based on the optimal threshold identified for adults to dichotomize the continuous AI-output (0-100). The green diamonds show optimized cut-offs calculated for maximizing the sum of sensitivity and specificity.. The performance metrics based on adult threshold of 15 (blue) and optimized cutoffs for children (green) are shown in the column on the right side (sens = sensitivity, spec = specificity, PPV = positive predictive value, NPV = negative predictive value).

population in pediatric patients was already investigated by Morcos at., who explored the reliability of an algorithm from an open source library for chest radiograph datasets (TorchXRayVision) for pneumonia detection ([13,23]. The authors found a decent but substantially lower performance compared to our study (sensitivity 79.83%, specificity 67.66%, PPV 86.95%, NPV 55.41%). The authors highlighted that although models exclusively trained on pediatric images would likely perform better, AI research on pediatric chest radiograph interpretation could be expedited by leveraging adult-based algorithms.

A similar approach as in the presented study was previously reported by Shin et. al., who investigated the diagnostic performance of the same AI-tool in a pediatric cohort (0–18 years) [12]. Exclusion of children under the age of 2 years and cardiomegaly increased the accuracy to as high as 96.9%, comparable to the performance for adult patients. In keeping with the findings of our study, they observed that age of patients with incorrect diagnoses was significantly younger than those with correct diagnosis (median 1 year vs. 7 years, p < 0.001). Moreover, age emerged as a significant factor for incorrect diagnosis in the logistic regression test. While noting a better performance of the tool in this study, compared to our results, it is crucial to note a fundamental distinction in the study design. We conducted a dedicated reading session informed by clinical information and radiological expertise, followed by a separate assessment of the AI-tools performance to avoid any potential bias. In contrast, in the above mentioned study, the radiologist was presented with the AI-output and could then confirm the abnormalities in reference to the results of the AI-tool, which was considered as a reference read. Nonetheless, these results emphasize the need for a refined approach: while older children could benefit from an extended application of adult chest radiograph algorithms, simultaneous developmental efforts are needed for younger subjects. This is underscored by our analysis calculating optimized thresholds to binarize the AI-output, which, without any retraining of the AI-tool, allowed for a performance increase, especially in the patients aged 2–6 years. Although we refrained from pathology specific cut-offs due to a small sample size of individual pathologies, this additional step highlights the potential to further refine the diagnostic performance of the AI tool by calculating specific cut-offs for children in larger data sets. One such approach was used in another study where separate operating points were calculated for children based on lesion type, age and imaging method in a larger dataset to improve the diagnostic performance of the AI tool [11].

The spectrum of pathologies differs significantly in pediatric patients. In our study, we noted that in cases of rare pathologies, such as sequestration, Ewings-Sarcoma of the first rib, and a chest wall lymphangioma producing a pleural shadow; the algorithm demonstrated reasonable sensitivity in detection even though the classification was inaccurate, which is partly explained by overlapping radiological findings of various clinical entities (Fig 3). Furthermore, the

performance of the AI tool cannot be extrapolated to certain respiratory conditions like tuberculosis, where radiological manifestation of TB are different among the pediatric and adult populations [24]. This further highlights the possible role of AI-algorithms in abnormality detection while leaving the role of interpretation to an expert for this patient population.

Our study has the following limitations. Firstly, though the sample was enrolled from clinical routine, the frequency of pathologies was relatively small in the study population. Secondly, although the newly defined optimized thresholds for children allowed for a performance increase, especially in younger individuals, further validation in larger and more diverse datasets is necessary to test for generalizability. Our study was conducted in a predominantly European pediatric population, reflecting the demographic composition of the study region. Future studies involving more ethnically diverse cohorts are warranted to assess the generalizability and fairness of AI applications in pediatric imaging, and to account for potential algorithmic biases related to ethnicity [25]. Finally, even though care was taken to generate high quality reference reads, minor variances in image interpretation such as missing/over diagnosing slight pulmonary edema or atelectasis could not be avoided. This could have been partially be off-set with a formal multi-reader consensus panel for ground truth reading to improve the replication of radiological analysis. Furthermore, during a focused review of cases to identify potential false positives, we identified a few cases such a discrete atelectasis or borderline cardiomegaly, which were flagged by the AI tool but not marked in the reference reads. While it is also important to note that AI may "overdiagnose" subtle findings that experienced radiologists would reasonably judge as clinically insignificant or not warranting formal reporting, this could suggest a potential role of AI tools in highlighting findings that could be overlooked in the clinical routine.

In conclusion, a repurposed AI tool developed for adult chest radiograph diagnosis but applied to a pediatric population showed high and clinically acceptable diagnostic performance for relevant findings in this independent validation study. Additional fine-tuning may help to further increase performance and support clinical decision-making in a particularly vulnerable patient population.

## Supporting information

**S1 Table. Results of the algorithm performance in children aged 2–6 years.**
(DOCX)

**S2 Table. Results of the algorithm performance in children aged 7–14 years.**
(DOCX)

**S1 Fig. Flow charts showing the performance for all the relevant pathologies (a) in all age groups, (b) 7–14 years, and (c) 2–6 years. A)** All relevant pathologies. **B)** Pleuroparenchymal pathologies. **C)** Mediastinal pathologies.
(TIF)

**S2 Fig. Definition of optimal thresholds for the performance of the AI-tool in children.** The dotted blue line represents the pre-defined vendor recommended threshold of 15, which is based on the optimal threshold identified for adults to dichotomize the continuous AI-output (0–100). The green diamonds show optimized cut-offs calculated for maximizing the sum of sensitivity and specificity. The performance metrics based on adult threshold of 15 (blue) and optimized cutoffs for children (green) are shown in the column on the right side (sens = sensitivity, spec = specificity, PPV = positive predictive value, NPV = negative predictive value)
(TIF)

## Author contributions

**Conceptualization:** Prerana Agarwal, Alexander Rau, Fabian Bamberg, Elmar Kotter, Jakob Weiss.

**Data curation:** Prerana Agarwal, Helen Ngo.

**Formal analysis:** Prerana Agarwal.

**Investigation:** Prerana Agarwal.

**Methodology:** Prerana Agarwal, Jakob Weiss.

**Project administration:** Alexander Rau, Jakob Weiss.

**Resources:** Alexander Rau, Ambika Seth.

**Software:** Ambika Seth.

**Supervision:** Alexander Rau, Fabian Bamberg, Elmar Kotter.

**Writing – original draft:** Prerana Agarwal, Jakob Weiss.

**Writing – review & editing:** Prerana Agarwal, Alexander Rau, Helen Ngo, Fabian Bamberg, Elmar Kotter, Jakob Weiss.

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
