## [Decision Letter · Decision Letter 0]

PONE-D-25-04141Deep Learning for Pediatric Chest X-Ray Diagnosis: Repurposing a Commercial Tool Developed for AdultsPLOS ONE

Dear Dr. Agarwal,

Thank you for submitting your manuscript to PLOS ONE. After careful consideration, we feel that it has merit but does not fully meet PLOS ONE’s publication criteria as it currently stands. Therefore, we invite you to submit a revised version of the manuscript that addresses the points raised during the review process.

Thank you for submitting this interesting manuscript to our journal. The use of AI is the most talked about thing in healthcare industry right now and we highly appreciate your work in this field. There is also notable gaps in evidence base for its use case in childhood TB. We are very happy to inform you that this paper has been reviewed by relevant experts and I am also very happy to see the time and effort they have invested in this review. I believe that addressing their comments/feedback will significantly improve the scientific validity of the manuscript. We look forward to the revised submission. 

We look forward to receiving your revised manuscript.

Kind regards,

Shahriar Ahmed, MBBS, MHE, MPhil

Academic Editor

PLOS ONE

Journal Requirements:

“Our institute received a grant from Lunit for technical support of the study”

5. We note that you have indicated that there are restrictions to data sharing for this study. PLOS only allows data to be available upon request if there are legal or ethical restrictions on sharing data publicly. For more information on unacceptable data access restrictions, please see http://journals.plos.org/plosone/s/data-availability#loc-unacceptable-data-access-restrictions.

Additional Editor Comments:

Thank you for submitting this interesting manuscript to our journal. The use of AI is the most talked about thing in healthcare industry right now and we highly appreciate your work in this field. There is also notable gaps in evidence base for its use case in childhood TB. We are very happy to inform you that this paper has been reviewed by relevant experts and I am also very happy to see the time and effort they have invested in this review. I believe that addressing their comments/feedback will significantly improve the scientific validity of the manuscript. We look forward to the revised submission.

Reviewers' comments:

Reviewer's Responses to Questions

**Comments to the Author**

1. Is the manuscript technically sound, and do the data support the conclusions?

Reviewer #1: Yes

Reviewer #2: Yes

2. Has the statistical analysis been performed appropriately and rigorously? 

Reviewer #1: Yes

Reviewer #2: Yes

3. Have the authors made all data underlying the findings in their manuscript fully available?

Reviewer #1: No

Reviewer #2: No

4. Is the manuscript presented in an intelligible fashion and written in standard English?

Reviewer #1: Yes

Reviewer #2: Yes

5. Review Comments to the Author

Reviewer #1: This is important research that aims to advance the performance of AI CXR models among pediatric patients, a patient group that has not received attention in the digital X-ray space despite the accelerated development of this field for adults. Authors assessed how a CAD algorithm that was previously trained on adults performs among children and found that performance was clinically acceptable, though further cutoff tuning could help further improve.

I have a few comments:

Abstract:

• I recommend naming the commercial tool directly in the abstract so that researchers interested in this tool can identify your paper more quickly.

• Would mention out of the 958 how many had abnormal CXR (n=200) for a bit more context on the study population

• “The diagnostic performance of the AI-tool was validated using standard measures of accuracy using recommended and optimized thresholds to dichotomize the continuous AI output (0-100)” – clarify that the recommended thresholds are for adults.

• I am not a proponent of reporting on the accuracy of an algorithm. Accuracy can be highly misleading in imbalanced datasets as is the case with only 200 positives. For overall performance, I would report the AUC and then the sensitivity/specificity, specifying that this is against the adult-recommended threshold. For the age-stratified results, I would only report the AUCs in the abstract.

Introduction

• Again, would name the actual CXR device here. You don’t introduce what the tool is until the methods.

Methods

• The initials of the radiologists are still “XX”

• Inconsistent nomenclature for figure (“Fig.” vs “Figures”)

• “The following ten findings are detected by the algorithm: consolidation, atelectasis, nodule, fibrosis, calcification, pleural effusion, pneumothorax, pneumoperitoneum, cardiomegaly and mediastinal widening.” – could you provide the N for each of these that were used when training the model on the adult population. Would allow for a comparison with the pediatric cohort.

• You are missing the diagnostic performance metrics (AUC, sens, spec…) of the CAD in the adult population. I see it in figure 5, but this could be missed and should be in the main text to allow for comparison with what is achieved in pediatrics.

Results

• When discussing threshold-based metrics (accuracy, sens, spec, PPV, NPV) it is good to remind the reader that this is benchmarked against the adult threshold.

• The corrected analysis is interesting to highlight the imperfect reference standard (i.e., human radiologist). I think it highlights a limitation of the study that is not acknowledged in the discussion, that there was only one board-certified radiologist reviewing the X-rays compared to the 5 used in the adult study. This is evidenced by the fact that 6/7 corrected scores were actually when the AI was right but the labeller was wrong. I am not sure if re-calculating the diagnostic metrics for the corrected analysis makes sense as this is altering the definition of what the reference actually is. It may be sufficient to descriptively highlight these discrepant cases, and mention this lack of replication of radiological analysis as a limitation in the discussion.

• “we calculated optimized cut-offs for each pathology and their performance metrics” – name the two pathology groups because I initially thought it was for each individual pathology (which is underpowered). It would also be good to have an overall re-calculated thresholds for any abnormal vs normal for pediatric, as was done for adults.

Discussion

• As previously mentioned, acknowledgement of a lack of replication of radiological interpretation in the reference needs to be acknolweged

• Further limitations are that performance cannot be extrapolated to other respiratory conditions (e.g., tuberculosis CAD X-ray interpretation in adults vs in pediatrics cannot draw conclusions from this study). Could go along with the generalizability statement already made.

Tables

• Table 1: the pleuroparenchymal and mediastinal denominator should be 200. Would also be good to show the number who have both as a separate line .

• Inconsistent N for the relevant pathology, pleuro, mediastinal across table 1, 2a and 2b. I expect the number in 1 to match 2a since 2b is the correct. However relevant overall in 1 is N=200 while in 2a is N=198, etc.

• Figure 1: there is a lot of text, would be good to cut down. No definition of what (a) and (b) represent. Would be good to also show performance metrics for both adults and pediatrics so that it gives all the info in one place.

• Figure 2: doesn’t add much to the manuscript, can be deleted.

Reviewer #2: Dear Editor, authors,

I am grateful for the opportunity to review the manuscript “Deep Learning for Pediatric Chest X-Ray Diagnosis: Repurposing a Commercial Tool Developed for Adults”. The authors note that there are few dedicated AI tools for evaluating pediatric chest radiographs. They, therefore, set out to study how a tool developed for the analysis of chest radiographs of adults performs when used (“repurposed”) for pediatric chest radiographs.

Title:

Fine.

Keywords:

Please revisit the keywords. My suggestion: Pediatric; Chest Radiograph; Artificial Intelligence

Abstract:

Overall informative. Please consider the following suggestions:

Rare  largely unavailable; Any discordant findings  All discordant findings

The authors state: “The diagnostic performance of the AI-tool was validated using standard measures of accuracy using recommended and optimized thresholds to dichotomize the continuous AI output (0-100).” I think some readers might struggle with this and therefore ask the authors to elaborate on this.

The performance was high for relevant findings. Please elaborate what relevant means in this context.

The readers may find the categorization according to the 7-year cut-off somewhat arbitrary. The authors may want to touch on this point, too.

Competing Interests:

The authors have declared that no competing interests exist. With all due respect, this statement must be further justified given that the developer of the studied software tool supported the study and that one of the authors is affiliated with the company itself. I also notice that the authors report on page 17 that the developer financed two of the key authors through their institution. I warmly welcome full transparency. If possible, consider stating that although the developer funded the study and one of the authors is employed by the company, all authors had access to the data and the decision to publish was based on the findings, not by the request of the company.

Ethics Statement:

Please also explicitly state the need for written informed consent was waived by the ethics committee and the study adheres to the national laws and regulations, if applicable.

Abbreviations:

I note the authors use the terms chest radiograph and chest X-ray. Please stick to chest radiograph through and through.

Introduction:

Although the introduction is well-written and reasoned, I think a more thorough review of existing tools for pediatric chest radiographs is very much needed. I therefore suggest well-conducted literature search and addition of a table summarizing the key results.

Materials and Methods:

Patient population:

The authors should describe the study as a single-center retrospective cohort study.

Perhaps the authors would like to explain why they chose to include 1000 consecutive chest radiographs. What led them to think this is an appropriate size for the cohort, not for example 900 or 1100 chest radiographs?

I think it’s reasonable to omit the exclusion criteria 2 because the authors already stated they included radiographs of older children. Rather, please reason why the study was limited to children aged between 2-14 years.

Ethics statement: please make sure it’s concordant with the ethics statement as discussed earlier.

Reference standard reading:

Please consider the following suggestions:

Fig. 3  Figure 3.

Please explain why all the available information, including the lab results, CT scans, and clinical history were taken into consideration when laying out the reference reads. After all, this information was not available to the software. While I of course support the use of all available information when reading the radiographs, one might argue that a truer performance assessment could have been achieved without considering other information.

Chest radiographs:

Were the chest radiographs caputed in supine, standing or sitting position?

AI-Algorithm:

Rather than speaking of gender, please speak of sex. Men  males, women  females.

Was Lunit involved in the decision to publish the results?

Could you please also touch on the training set used in the validation study. I think it would be very important to discuss the homogeneity/heterogeneity of the dataset particularly in relation to sex, age and ethnicity of the patients.

Statistical Analysis:

I wish to applaud for the well-written chapter. Normally this section is overlooked. I find it easy to immerse into the Results section after reading this chapter. However, I ask the authors to elaborate on the age categorization.

Results:

All in all, this section is well written and nicely backed by the tables. However, as the tables should stand alone, I think the authors could think of further strengthening the tables by informing the reader where the numbers come from. For example, it should be clear what is being used as the gold standard and why the number of relevant overall pathologies differ between the tables.

Please also report the results for girls/boys separately in Table 1.

I’d also encourage the authors to report the ethnicities of the patients.

Discussion:

As I suggested earlier, I prompt a more thorough review and discussion of the existing tools for pediatric chest radiograph assessment.

I’d like the authors to also discuss whether the results would translate to all demographic subgroups of children. Particularly, does ethnicity have an effect?

Table 2b:

“a simplified manner to avoid bias due to low number of cases.” � please rewrite for clarity.

Figure 1:

Please also indicate the study setting: this is a single-center retrospective cohort study.

It said in the results 200 (20.9%) of the chest radiographs had at least one pathology. Figure 1 seems to suggest there were 198 chest radiographs with abnormal findings.

Of note, it seems the fonts are widely inconsistent between the Figures.

Figure 2:

Please name the Figure descriptively keeping in mind that the Figure and the description should be able to “stand alone”, i.e., be clear without reading the text.

If the name of the title refers to the CONSORT reporting system, the word should be fully capitalized.

Please correct the inconsistent spacings.

Figure 3:

There seems to be some kind of an annotation in the upper right corner of the A part. Was this intentional?

Figure 4:

OK, very informative.

Figure 5:

Not sure if the figure is needed. I would not oppose its inclusion, however, I feel it does not really add too much information. Should the authors want to keep it, I’d ask them to avoid abbreviations sens, PPV, NOV and spec.

6. PLOS authors have the option to publish the peer review history of their article (what does this mean? ). If published, this will include your full peer review and any attached files.

**Do you want your identity to be public for this peer review?** For information about this choice, including consent withdrawal, please see our Privacy Policy .

Reviewer #1: No

Reviewer #2: No

---

## [Author Response · Author response to Decision Letter 1]

25 May 2025

Point-by-Point Response to Reviewer Comments

PONE-D-25-04141

Deep Learning for Pediatric Chest Radiograph Diagnosis: Repurposing a Commercial Tool Developed for Adults

General Response: We thank the editor and reviewers for their thoughtful comments and have addressed all concerns in the revised version of the manuscript. We have also formatted the manuscript to suit the journal´s style and reorganized the layout to include the tables and figure texts in the manuscript. A point-by-point response to reviewers is provided below. The page and line numbers refer to the manuscript version with tracked changes.

Response to Reviewer 1

We thank the reviewer for the critical evaluation of our work and are grateful for their very helpful comments and suggestions that we have incorporated in our revised manuscript as detailed below:

Comment 1:

I recommend naming the commercial tool directly in the abstract so that researchers interested in this tool can identify your paper more quickly.

Response 1:

We have added the name of the tool to the abstract. (Page: 3, Line: 5)

Comment 2:

Would mention out of the 958 how many had abnormal CXR (n=200) for a bit more context on the study population.

Response 2:

We thank the reviewer for this suggestion. We have added the following sentence to the Abstract to provide this context (Page: 3, Line: 17):

“200 radiographs [20.9%] demonstrated at least one relevant pathology”.

Comment 3:

The diagnostic performance of the AI-tool was validated using standard measures of accuracy using recommended and optimized thresholds to dichotomize the continuous AI output (0-100)” – clarify that the recommended thresholds are for adults.

Response 3:

The sentence was changed as follows for clarity (Page: 3, Lines: 12-14 ):

“For this, the continuous AI output (ranging from 0-100) was binarized using vendor recommended thresholds recommended for adults and optimized thresholds identified for children.”

Comment 4:

I am not a proponent of reporting on the accuracy of an algorithm. Accuracy can be highly misleading in imbalanced datasets as is the case with only 200 positives. For overall performance, I would report the AUC and then the sensitivity/specificity, specifying that this is against the adult-recommended threshold. For the age-stratified results, I would only report the AUCs in the abstract.

Response 4:

Thank you for your valuable feedback.

We have changed the abstract as follows (Page: 3, Lines: 17-23)

“Using the adult threshold, the AI-tool showed a high performance for all relevant findings with an AUC 0.94 (95% CI: 0.92-0.95) and. In stratified analysis by age (2-7 vs. 7-14-years-old) a significantly higher performance (p<0.001) was found for older children with an AUC of 0.96 (95% CI: 0.94-0.98) with a sensitivity and specificity of 87.5% and 82.3% respectively, which further increased using optimized thresholds for children.”

Comment 5:

Again, would name the actual CXR device here (in introduction).

Response 5:

We have made the following changes to the introduction: Page: 5, Line: 2

“Here, we investigated the diagnostic performance of a commercially available AI-tool developed for adult chest radiograph interpretation (Lunit INSIGHT CXR) in a real-world clinical dataset of children aged 2-14 years old.”

Comment 6:

The initials of the radiologists are still “XX”

Response 6:

The initials were changed to “PA” (Page: 7, Line: 12)

Comment 7:

Inconsistent nomenclature for figure (“Fig.” vs “Figures”)

Response 7:

We changed “Fig.” and “Figures” to “Fig” to comply with the journal´s formatting guidelines.

Comment 8:

“The following ten findings are detected by the algorithm: consolidation, atelectasis, nodule, fibrosis, calcification, pleural effusion, pneumothorax, pneumoperitoneum, cardiomegaly and mediastinal widening.” – could you provide the N for each of these that were used when training the model on the adult population. Would allow for a comparison with the pediatric cohort.

Response 8:

We would like to thank the reviewer for this question. We have contacted Lunit for this question and have included their response here:

“While we are unable to disclose the exact number of annotated training cases for each individual finding due to proprietary limitations, I can confirm that Lunit INSIGHT CXR was trained on a dataset of approximately 280,000 chest X-rays with annotation and the algorithm was specifically trained to detect the following ten key findings: Consolidation, atelectasis, nodule, fibrosis, calcification, pleural effusion, pneumothorax, pneumoperitoneum, cardiomegaly, and mediastinal widening. All images used for training were confirmed by one of the following methods: Original radiology report + pathology confirmation or radiology report + CT confirmation, or independent image review by at least one expert radiologist. This approach ensured high-quality ground truth labeling for accurate model training. The dataset consisted of PA and AP images from adult patients, with data sourced from multiple countries and acquired using equipment from 24+ X-ray device manufacturers, providing significant diversity across pathologies and imaging conditions.”

Comment 9:

You are missing the diagnostic performance metrics (AUC, sens, spec…) of the CAD in the adult population. I see it in figure 5, but this could be missed and should be in the main text to allow for comparison with what is achieved in pediatrics.

Response 9:

We thank the reviewer for this comment. We would like to clarify that the diagnostic performance metrics shown in Figure 5 reflect the performance of the AI tool in the pediatric population, using the recommended threshold of 15, which is based on the optimal threshold identified for adults. To improve clarity, we have now explicitly stated this in the main text and the figure. We would like to highlight that performance metrics for the adult population were not the focus of this analysis.

Page 14, Lines: 12 -23

“The vendor-recommended threshold of 15 to binarize the continuous AI output (ranging from 0-100) is based on the optimal threshold identified for adults. Therefore, as a final step in our analysis, we calculated optimized cut-offs for children for the entire cohort. Overall, the optimal cut-off for the pediatric age group was found to be 44.5. This adjusted threshold yielded a similar AUC of 0.93 with an accuracy of 89.8% (95% CI: 87.7–91.6%), sensitivity of 80.0%, and specificity of 92.4%. Compared to the adult threshold, the pediatric-specific cut-off led to improvement in the specificity and the PPV, supporting a safer AI interpretation in children with a slight trade-off in the sensitivity at 80%. The sub-group specific thresholds for pleuroparenchymal and mediastinal subgroups are shown in the Fig 4 and for each pathology are shown in S1 Fig.

Page 14, Lines: 25-32

“Fig 5: Definition of optimal thresholds for the performance of the AI-tool in children. The dotted blue line represents the pre-defined vendor recommended threshold of 15, which is based on the optimal threshold identified for adults to dichotomize the continuous AI-output (0-100). The green diamonds show optimized cut-offs calculated for maximizing the sum of sensitivity and specificity for pleuroparenchymal and mediastinal subgroups in the entire cohort. The performance metrics based on adult threshold of 15 (blue) and optimized cutoffs for children (green) are shown in the column on the right side.”

Comment 10:

When discussing threshold-based metrics (accuracy, sens, spec, PPV, NPV) it is good to remind the reader that this is benchmarked against the adult threshold.

Response 10:

We thank the reviewer for this suggestion. Accordingly, we added the following sentence to Statistical analysis:

Page: 9, Lines 23-24

“In addition, sensitivity, specificity, positive predictive value (PPV), negative predictive value (NPV) were calculated after binarizing the continuous AI output using a predefined, vendor recommended threshold cut-off of 15, which was the optimal threshold identified for adults.”

We also changed the subheading in the results section as follows:

Page: 11, Line 9

“Performance analysis based on the thresholds recommended for adults:”

Comment 11:

The corrected analysis is interesting to highlight the imperfect reference standard (i.e., human radiologist). I think it highlights a limitation of the study that is not acknowledged in the discussion, that there was only one board-certified radiologist reviewing the X-rays compared to the 5 used in the adult study. This is evidenced by the fact that 6/7 corrected scores were actually when the AI was right but the labeller was wrong. I am not sure if re-calculating the diagnostic metrics for the corrected analysis makes sense as this is altering the definition of what the reference actually is. It may be sufficient to descriptively highlight these discrepant cases, and mention this lack of replication of radiological analysis as a limitation in the discussion.

Response 11:

We would like to thank the reviewer for this important observation. While one board-certified radiologist performed the reference readings, we would like to clarify that the original signed reports by pediatric radiologists were considered as second readings. Discrepant cases were subsequently reviewed and resolved through a consensus reading to establish the final reference standard for this study. We have acknowledged the limitation of not using a formal multi-reader consensus panel, in the Discussion section (see below)

Regarding the corrected analysis, we agree with the reviewer that recalculating diagnostic metrics based on retrospective case by case comparison could change the definition of the reference standard. We have therefore chosen not to include recalculated metrics and have thus omitted the paragraph “corrected performance analysis”. In keeping with this, we have removed the term “crude” performance analysis from the previous sections and have deleted the figure 2b. We now report these discrepancies descriptively in the Discussion.

Page: 17, Lines: 12-20

“This could have been partially be off-set with a formal multi-reader consensus panel for ground truth reading to improve the replication of radiological analysis. Furthermore, during a focused review of cases to identify potential false positives, we identified a few cases such a discrete atelectasis or borderline cardiomegaly, which were flagged by the AI tool but not marked in the reference reads. While it is also important to note that AI may “overdiagnose” subtle findings that experienced radiologists would reasonably judge as clinically insignificant or not warranting formal reporting, this could suggest a potential role of AI tools in highlighting findings that could be overlooked in the clinical routine.”

Comment 12:

“we calculated optimized cut-offs for each pathology and their performance metrics” – name the two pathology groups because I initially thought it was for each individual pathology (which is underpowered).

Response 12:

We would like to thank the reviewer for bringing this to our notice. We have changed the sentences as follows: (Page: 14, Lines: 21-23)

“The sub-group specific thresholds for pleuroparenchymal and mediastinal subgroups are shown in the Fig 4 and for each pathology are shown in S 1 Fig.”

Comment 13:

It would also be good to have an overall re-calculated thresholds for any abnormal vs normal for pediatric, as was done for adults.

Response 13:

We would like to thank the reviewer for this comment. We have now calculated the overall optimal threshold for children (Page:14, Lines: 17-21).

“Overall, the optimal cut-off for the pediatric age group was found to be 44.5. This adjusted threshold yielded a similar AUC of 0.93 with an accuracy of 89.8% (95% CI: 87.7–91.6%), sensitivity of 80.0%, and specificity of 92.4%. Compared to the adult threshold, the pediatric-specific cut-off led to improvement in the specificity and the PPV, supporting a safer AI interpretation in children with a slight trade-off in the sensitivity at 80%.”

We have updated the figure 4 accordingly (Page: 26)

Comment 14:

As previously mentioned, acknowledgement of a lack of replication of radiological interpretation in the reference needs to be acknowledged.

Response 14:

We thank the reviewer for highlighting this point. The following changes were made to the discussion. (Page: 17, Lines: 9-14)

“Finally, even though care was taken to generate high quality reference reads, minor variances in image interpretation such as missing/overdiagnosing slight pulmonary edema or atelectasis could not be avoided. This could have been partially be off-set with a formal multi-reader consensus panel for ground truth reading to improve the replication of radiological analysis.”

Comment 15:

Further limitations are that performance cannot be extrapolated to other respiratory conditions (e.g., tuberculosis CAD X-ray interpretation in adults vs in pediatrics cannot draw conclusions from this study). Could go along with the generalizability statement already made.

Response 15:

Thank you for the valid comment highlighting another important difference in disease manifestation in the pediatric population. We have added the following sentence to discussion (Page:16, Lines: 30-32):

“Furthermore, the performance of the AI tool cannot be extrapolated to certain respiratory conditions like tuberculosis, where radiological manifestation of TB are different among the pediatric and adult populations”

Comment 16:

Table 1: the pleuroparenchymal and mediastinal denominator should be 200. Would also be good to show the number who have both as a separate line .

Response 16:

We thank the reviewer for this point. We have updated the table 1 accordingly (Page: 17)

Parameter Total

(n = 958) Boys

(n = 524) Girls

(n = 434)

Age (years) * 6.75 ± 3.6 6.83 ± 3.6 6.65 ± 3.6

Projection (AP) 500/958 (52.2 %) 274/524 (52.3%) 226/434 (52.1%)

Relevant overall pathology 200/958 (20.9 %) 96/524 (18.3%) 104 /434 (23.4%)

• Pleuroparenchymal pathology † 162/200 (81.0 %) 83/200 (41.5 %) 79/200 (39.5 %)

• Mediastinal pathology ‡ 77/200 (38.5 %) 29/200 (14.5 %) 48/200 (24.0 %)

• Both mediastinal and pleuroparenchymal pathology 39/200 (19.5 %) 16/200 (8.0 %) 23/200 (11.5 %)

Comment 17:

Inconsistent N for the relevant pathology, pleuro, mediastinal across table 1, 2a and 2b. I expect the number in 1 to match 2a since 2b is the correct. However relevant overall in 1 is N=200 while in 2a is N=198, etc.

Response 17:

We would like to thank the reviewer for this and would like to apologize for this mistake. The values have been corrected (please refer to response 16).

Comment 18:

Figure 1: there is a lot of text, would be good to cut down. No definition of what (a) and (b) represent. Would be good to also show performance metrics for both adults and pediatrics so that it gives all the info in one place.

Response 18:

We thank the reviewer for this comment. The figure has been adjusted accordingly. The figure description was also changed to reflect the definition of (a) and (b). Since we have only evaluated the diagnostic performance in the pediatric population, we have only depicted the study methology in figure 1 and not included results. (Page: 6, Lines: 9-15)

Fig 1: Brief summary highlighting the study methology. a) AI Tool Development and Pediatric Repurposing: The AI tool was originally trained and validated using a large dataset of adult chest radiographs. For pediatric validation, the tool was retrospectively tested on 958 pediatric chest radiographs (CXR) from children aged 2–14 years.

b) Diagnostic Performance Analysis: The AI tool’s diagnostic performance in children was assessed using vendor-recommended thresholds, stratified by age groups (2–6 and 7–14 years), and optimized pediatric-specific thresholds.

Comment 19:

Figure 2: doesn’t add much to the manuscript, can be deleted.

Response 19:

We thank the reviewer for this feedback and have removed Figure 2. The numbering of other figures has been changed accordingly.

Response to Reviewer 2

We thank the reviewer for the critical evaluation of our work and are grateful for the

---

## [Decision Letter · Decision Letter 1]

Deep Learning for Pediatric Chest X-Ray Diagnosis: Repurposing a Commercial Tool Developed for Adults

PONE-D-25-04141R1

Dear Dr. Agarwal,

We’re pleased to inform you that your manuscript has been judged scientifically suitable for publication and will be formally accepted for publication once it meets all outstanding technical requirements.

Kind regards,

Shahriar Ahmed, MBBS, MHE, MPhil

Academic Editor

PLOS ONE

Additional Editor Comments (optional):

We would like to congratulate the authors for successfully addressing all comments raised by the reviewers. Thank you for considering this journal for publishing your important work. We hope that you will consider us again for your future publications.

Reviewers' comments:

Reviewer's Responses to Questions

**Comments to the Author**

1. If the authors have adequately addressed your comments raised in a previous round of review and you feel that this manuscript is now acceptable for publication, you may indicate that here to bypass the “Comments to the Author” section, enter your conflict of interest statement in the “Confidential to Editor” section, and submit your "Accept" recommendation.

Reviewer #1: All comments have been addressed

Reviewer #2: All comments have been addressed

2. Is the manuscript technically sound, and do the data support the conclusions?

Reviewer #1: Yes

Reviewer #2: Yes

3. Has the statistical analysis been performed appropriately and rigorously? 

Reviewer #1: Yes

Reviewer #2: Yes

4. Have the authors made all data underlying the findings in their manuscript fully available?

Reviewer #1: No

Reviewer #2: No

5. Is the manuscript presented in an intelligible fashion and written in standard English?

Reviewer #1: Yes

Reviewer #2: Yes

6. Review Comments to the Author

Reviewer #1: (No Response)

Reviewer #2: Thank you for addressing the comments. I am happy with the revisions and support the publication of this interesting manuscript.

7. PLOS authors have the option to publish the peer review history of their article (what does this mean? ). If published, this will include your full peer review and any attached files.

**Do you want your identity to be public for this peer review?** For information about this choice, including consent withdrawal, please see our Privacy Policy .

Reviewer #1: No

Reviewer #2: No

---

## [Editor Report · Acceptance letter]

PONE-D-25-04141R1

PLOS ONE

Dear Dr. Agarwal,

I'm pleased to inform you that your manuscript has been deemed suitable for publication in PLOS ONE. Congratulations! Your manuscript is now being handed over to our production team.

Kind regards,

on behalf of

Dr. Shahriar Ahmed

Academic Editor

PLOS ONE